# Medicinal Plants and Their Traditional Uses in Local Communities around Cherangani Hills, Western Kenya

**DOI:** 10.3390/plants9030331

**Published:** 2020-03-05

**Authors:** Yuvenalis M. Mbuni, Shengwei Wang, Brian N. Mwangi, Ndungu J. Mbari, Paul M. Musili, Nyamolo O. Walter, Guangwan Hu, Yadong Zhou, Qingfeng Wang

**Affiliations:** 1Key Laboratory of Plant Germplasm Enhancement and Specialty Agriculture, Wuhan Botanical Garden, Chinese Academy of Sciences, Wuhan 430074, China; yuvemorara@yahoo.com (Y.M.M.); wangshengwei@wbgcas.cn (S.W.); briannjoroge62@gmail.com (B.N.M.); john.mbari@students.jkuat.ac.ke (N.J.M.); guangwanhu@wbgcas.cn (G.H.); 2University of Chinese Academy of Sciences, Beijing 100049, China; 3National Museums of Kenya, East African Herbarium, P. O. Box 45166, Nairobi 00100, Kenya; pmutuku@museums.or.ke (P.M.M.); wuonyamolo@yahoo.com (N.O.W.); 4Sino-Africa Joint Research Center (SAJOREC), Chinese Academy of Sciences, Wuhan 430074, China; 5Center of Conservation Biology, Core Botanical Gardens, Chinese Academy of Sciences, Wuhan 430074, China

**Keywords:** Cherangani Hills, ethnobotanical, medicinal plants, decoction, informants, ailment categories, conservation

## Abstract

Medicinal plants are vital sources of easily accessible remedy used in the countryside healthcare system. This study aimed to find and make record of plants that are used for medicinal therapy by three communities living in Cherangani Hills. So far no single study has documented medicinal plants as a whole in the area. Ethnobotanical data were obtained through interviewing informants using semi-structured questionnaires and extracting information from journals and books. Descriptive statistical analysis was applied to describe the data. Overall 296 plant species from 80 families and 191 genera were identified. Asteraceae family was the most dominant, representing 10.7% of the total plant species recorded. Roots (35.9%) represented the most commonly used parts of the plant. The commonly used method of preparation was decoction (54.9%). The reported diseases were classified into 14 diverse ailment groups out of the 81 health conditions on their underlying user reports. Rural communities in Cherangani Hills are rich sources of plants with medicinal properties. Therapeutic uses of the compiled plants provide basic information that can aid scientists to conduct additional research dedicated to conservation of species and pharmacological studies of species with the greatest significance.

## 1. Introduction

Medicinal plants have been a vital source of both curative and preventive medical therapy preparations for human beings, which also has been used for the extraction of important bioactive compounds [1,2,3]. It is estimated that almost 80% of the world’s total population, regularly, depends on traditional medicine and products for its healthcare needs especially in third world countries. Many sick people in the developing regions combine the conventional medicine with traditional medicine [4,5,6]. Traditional medicines are usually cheaper than modern medicines, and probably the only natural remedies available and accessible in the remote rural communities in developing countries [7]. Rural dwellers prefer traditional medicines because of their close proximity to the traditional healers and the fact that the healers understand their culture and environment as well as their patients. In rural areas, access to western healthcare is a problem especially in the Sub-Saharan countries, because conventional healthcare is concentrated in towns [8]. Plant medicine has continuously been practiced for a long period, especially in some African tribes with a long history [9]. The Kenyan diversified flora with over 7000 plant species is one of the richest in East Africa [10]. Consequently, the higher number of plant species have led to discovery of many medicinal plants in the region. In Kenya, more than 70% of the people use local home-made remedies as their first source of medicine, while more than 90% use plant related remedies at one time or another [11]. Phytotherapy is another fundamental part of the native communities of Kenya who have vital indigenous knowledge acquired through generations. However, this practice is often less transferred owing to industrialization and adoption of western life style. Traditional knowledge in many Kenyan ethnic tribes remain untapped since the medicinal plants have not been fully documented as the information is passed orally from one generation to the other posing danger of its loss [8,10].

Indiscriminate trade of plant resources, uncontrolled collecting methods, habitat change, overexploitation, and climate change pose great threats to availability of plant medicine in most third world countries, thus, creating a pressing need for better methods of conservation and viable use of priority plant resources [12]. In Kenya, research on ethnobotany has been on going after independence and several publication of guides and books have been published [13,14,15,16]. Recording and preserving the traditional knowledge on medicinal plants has become very important practice in recent times [17]. Several ethnobotanical and ethnopharmacological research studies have been published documenting Kenya’s medicinal plant knowledge and use: Marakwet county [11,18], Northern Kenya [19], Siaya county [20,21], Tugen [22], Machakos county [23,24], Samburu county [25,26,27], Sekanani Valley, Maasai Mara [28], Kajiado county [29,30,31,32,33], Embu and Mbeere county [34], Makueni county [35], Mount Elgon [36], Nakuru county [37], Nandi county [38,39,40,41], Tharaka Nithi county [42], Kakamega county [43,44,45,46,47], Kitui county [48], Elgeyo Marakwet county [49], Kericho county [50], Machakos county [51], Narok county [52,53,54], Trans-Mara county [55], Kilifi county [56]. However, in Kenya, many areas and ethnic societies are yet to be ethno botanically surveyed.

This study focused on three communities in Cherangani Hills and the medicinal plants used to treat different ailments. The documentation of the natural resources is key as it will assist in the conservation of residual and remaining forests [38]. The databases obtained in this research forms a foundation for potential development of new medicines [10]. Ethnobotanical investigations are vital in preserving traditional medicine through suitable documentation of plants, which also assist in its sustainability [7]. Previous studies have been carried out in sections of Cherangani hills hence this research aimed to cover medicinal plants in the entire study region.

## 2. Material and Methods

### 2.1. Study Area

This study covered the human settlement areas around and adjacent to the Cherangani Hills forest ecosystem found in the western side of Kenya (Figure 1). Cherangani Hills reserve (35°26′ E, 1°16′ N), cuts across three counties, namely Trans Nzoia (1551 Ha), Elgeyo-Marakwet (74,249 Ha), and West Pokot (34,380 Ha), totaling 110,181 Ha, and is occupied by three ethnic groups comprising Luhya, Marakwet, and Pokot people respectively. The hills comprise 12 forest blocks where medicinal plant resources were collected and include Kipkunurr, Kapolet, Sogotio, Chemurkoi, Kaisungor, Cheboyit, Embobut, Kererr, Kiptaberr, Kapkanyar, Toropket, and Lelan [57].

### 2.2. Selection of Respondents

Purposive sampling was applied in the field investigation, where traditional therapists and elders helped to pin point medicinal plant practitioners and emphasis was laid on both women and men [58,59]. Seventy-eight practitioners (38 women and 40 men) were sampled near each of the 12 forest block locations. Selected group of respondents were distinguished in the region because of their long tradition in providing services allied to traditional health remedies. Fifty-one practitioners were traditional healers and the remaining number were village elders who had acquired familiarity on medicinal healing skills of plants from their parents and close relatives.

### 2.3. Ethnobotanical Data Collection and Plant Identification

Ethnobotanical information were gathered between September 2017 and January 2019 by interviewing, using methodological ways designed in ethnopharmacological in field data collection. The local chiefs were informed afore about the initiation of the survey and permission was allowed. Interviews, discussions, formal and informal conversations, as well as field visits were conducted [60]. More information was sourced from literature studies including journal articles and books [61]. Botanical names, local names, diseases treated, method of preparation, dosage, and modes of administration were recorded. An interview was carried in the local dialect and translated to English. Data on habit, habitat, and plant parts used were recorded. For each described plant species, a specimen was taken and preliminary identification was performed in the field. The specimens were pressed, dried, and the identification results were confirmed at the East African herbarium. A specimen voucher number was given and prepared for each collected herbarium specimen and deposited in the East African Herbarium (Appendix A). Authentication of identified plant specimens was verified using the Flora of East African by comparisons with authenticated specimens at the East African Herbarium (EA), Nairobi, Kenya. The scientific names indicated in Appendix A, in this research work are the recognized names according to “The plant List” database.

### 2.4. Data Analysis

#### 2.4.1. Informant Consensus Factor

Informant consensus factor (ICF) was computed using a mathematical expression: ICF = (N_ur_ − N_t_)/(N_ur_ − 1), where N_ur_ refers to the summed up number of citations for each disease group and N_t_ is the number of plant species used in that category [62,63]. The lowest ICF value is 0.00 and the highest is 1.00. Low ICF values indicates that informants do not agree on which plant medicine to use in a particular ailment, while high ICF values indicate that a limited number of plant species are known to be administered by a large number of informants to treat a specific disease. High ICF values can further be investigated and used to find species of important bioactive compounds [64].

#### 2.4.2. Fidelity Level (FL)

Fidelity level (FL) is the total number of informants who referenced the consumption of some medicinal plants to treat a specific disease in the region and is calculated by the following formula: FL = Np/N × 100, where Np represents total number of informants citing the use of the plant to be administered to a particular disease and N denotes the total number of informants who utilized the plants as a medicine group [63,65]. Plant species with a higher percentage of FL shows the frequency and high usage in healing a specific disease by the informants in the community and vice versa when the percentage is low.

#### 2.4.3. Jaccard’s Coefficient of Similarity (JCS)

Jaccard’s coefficient of similarity (JCS) was computed to compare the medicinal plant composition and their similarity with other counties in Kenya. Similarity values were computed between other areas already studied by other researchers in different regions in comparison with the present study area. JCS, was calculated as: JCS = c/(a + b + c), a representing the total number of medicinal plant species obtained in area A, b is the total number of medicinal plant species discovered only in area B, and c is the total number of common plant species occurring in areas A and B [66].

## 3. Results

### 3.1. Demographic Profile of Respondents

A total of 40 (51.2%) males and 38 females (48.7%) were interviewed. The results between male and female informants were almost equal. The lowest age of informants was 15 and the highest 85 years, with the highest modal class being (66–75) years, representing 30.8%. The frequency of other age class include, 15–25 (1.3%), 26–35 (3.9%), 36–45 (6.4%), 46–55 (12.8%), 56–65 (19.2%), 66–75 (30.8%), >76–85 (25.6%) (Table 1). Illiterate (42.1%), Primary (33.3%), Secondary (21.8%), Tertiary (2.7%) (Table 1).

### 3.2. Diversity of Medicinal Plant Use

This study compiled 296 medicinal plants traditionally managing various human diseases (Appendix A) resulting to 80 families and 191 genera. The largest percentage of medicinal plants obtained belonged to the family Asteraceae (32 species), followed by Leguminosae (28), Lamiaceae (18), Rubiaceae (14), Euphorbiaceae (12), Apocynaceae (10), Malvaceae (10), and Anacardiaceae (8). The result revealed that species in Leguminosae family contained the highest percentage (8.7%) in treating different ailments. This was followed by Asteraceae (7.7%), Lamiaceae (6.1%), Rutaceae, Anacardiaceae (4.6% each), and with the rest of the families treated less than 4.2% of the ailments (Table 2).

### 3.3. The Habitat for Medicinal Plants

The most common plant habitat identified was bushland 20.0%, followed by escarpment 17.9%, highland forest 14.8%, grassland forest 13.8%, woodland 9.5%, riverine 7.4%, valley 6.2%, cultivated 4.8%, wooded grassland 2.9% and forest margins at 2%. Figure 2 constitutes the habitats of medicinal plant species, of Cherangani hills, consequently a high plant diversity for the production of roots, bark, leaves, fruits, and flowers as medicinal resources.

### 3.4. Habit, Parts Used for Medicine and Methods of Preparation

Growth habit of shrubs have the highest percentage of 35.1% of the total medicinal plants in this study. Total of 27.5% of trees are represented by the total number plant species (Figure 3a), followed by herbs (26.5%), climbers (10%), epiphytes, and parasites with 0.3% each. The plant parts used include roots (35.9%), leaves (34.9%), bark (15.0%), fruits (5.2%), branches (5.0%), whole plant (1.9%), flowers (1.1%), seeds (0.2%), and barks of roots (0.2%) (Figure 3b). People living in the study area use different methods to prepare different medicines for treatment of different ailments. Decoction (boiling) proved to be used more commonly as the mode of preparation (53.3%), followed by pounding/crushing (24.5%), and chewing (9.3%). Other preparation methods represented less than 5% (Figure 4).

### 3.5. Informant Consensus Factor (ICF)

To obtain the accurate ICF, the reported diseases were grouped into 14 different ailment groups out of the 81 health conditions based on their use reports (Table 3). The results of the reported ailments are as follows; digestive system disorders (25.2%), respiratory tract infections (18.3%), parasitic diseases (17.9%) (Table 3). Stomachache (8.9%), malaria (6.7%), aphrodisiac (6%), coughing (4.8%), and abdominal pains (3.4%) were the most common disease mentioned. Within the three major disease groups, digestive system disorders had 139 use-reports, followed by respiratory tract infections (101) and parasitic diseases and other infections (99) use-reports. The greatest ICF (0.79) was mostly for metabolic disorders, followed by gynecological issues (0.76). Respiratory tract infections, erectile dysfunctions, and impotence were less frequently and had the lowest IFC of 0.27 and 0.18 respectively.

### 3.6. Fidelity Level (FL)

The calculated fidelity level (FL) of 18 important plant species varied from 36.2 to 90.9% (Table 4). *Carissa spinarum* L. and *Warburgia ugandensis* Sprague depicted 90.5% and 90.9% FL respectively against malaria and respiratory disorders as the most utilized plants. *Asparagus racemosus* Willd. and *Tragia brevipes* Pax. registered FL of (36.2%) and (38.5%) treating kidney diseases and rheumatism respectively. *Clausena anisata* (Willd.) Hook.f. ex Benth. at 60% FL proved to treat heart diseases according to user reports in the study area. Respondents also preferred using *Basella alba* L. as a vegetable and in the study area it stood at 85.7% FL.

### 3.7. Jaccard’s Coefficient of Similarity

This study represents the first scientific documentation of ethnobotanical uses of 296 medicinal plant used by the three communities in Cherangani hills. The current report on the ethnomedicinal uses of plants was compared to those of previous studies done in other regions of Kenya (Table 5). It was found that Marakwet (18%), Sungurur (16%), and Keiyo (14%) had the highest Jaccard’s coefficient of similarity in the makeup of medicinal plant species whereas the degree of similarity was lower in areas like Nandi (0.05%) and Kitui (0.06%) (Table 5).

### 3.8. Threats to Medicinal Plants

Informants’ responses showed that many factors have contributed to the threats faced by plants of medicinal importance in the study area. (Table 6). Agricultural expansion (38.5%) was the main threat to important medicinal species, followed by overgrazing (20.5%), overharvesting (17.9%), firewood and Charcoal production (10.3%), environmental degradation (7.7%). Some respondents pointed out that other threats exist within the study area that are a result of deforestation and loss of habitat (5.1%).

## 4. Discussion

The communities around Cherangani hills forest reserve use a large diversity of flora in the treatment of a myriad of diseases and the native people have a broad traditional knowledge on plants of medicinal importance. The higher percentage of people that rely on medicinal plants could be attributed to the high cost of western medicine and inaccessibility of government medical facilities [33,74]. There was an insignificant difference between men and women in the knowledge of medicinal plants. Comparing with other study area in Kenya [51], there was no gender preference in the passing of medicinal plants knowledge from the parents to their offspring across local communities around Cherangani Hills. Informants in the age group above 45 years appeared to know more medicinal plants perhaps as a result of having more experience interacting with medicinal plants in their ecosystem. Additionally, fewer medicinal plants were known to those who attended tertiary levels of education compared to illiterate informants. The insignificant use of the plants of medicinal importance by the literates in the community can be attributed to lack of general preparation procedures and scientific information on their efficiency as well as their toxicity levels. Additionally, the collection as well as storage methods were identified as essential considerations by the literate members in the community. This indicates that there exists a generational disjunction in the passage of traditional medicinal plant knowledge. This can be linked to the influence of formal education as it was observed that illiterate informants had an upper hand in medicinal plants knowledge as compared to their tertiary level counterparts. Exposure of younger people to modern education and lifestyle has led them to prefer western medical treatment over traditional medicine hence despising medicinal plant treatments compared to those unexposed and uneducated [70].

Out of the 296 medicinal plants recorded, shrubs are commonly used because of their relatively higher resistance to drought, hence preferred as they are available for harvesting all year round [75]. Roots are most preferred compared to other parts of the plant as they are traditionally considered to have a higher strength of medicine and are readily available in all the seasons of the year [33,76,77,78]. Leaves are also highly utilized because they are obtained easily in large quantities in contrast to other plant parts. Moreover, a majority of traditional healers prefer to use leaves as they are considered to accumulate active ingredients by photosynthetic pigments such as alkaloids and tannins [79,80].

A myriad of methods of preparation are used within the three communities of Cherangani Hills and Kenya as a whole. In the study area it was uncovered that decoction was the most widely used method of preparation mainly because of the ease of using water to prepare them. Such a large variety of preparation methods that have been studied has been highlighted in some parts of Kenya and in other countries [36,48,51,61,76,81]. It has been established that more than one method is used in preparing many of the medicinal plants studied. However, the type of plant species, condition of ailment being treated, and plant parts used determined the method of preparation. Within the Marakwet, Luhya, and Pokot communities, the common way of administration of the prepared medicine was through drinking, which is in line with many other studies [36,48,51].

Gastrointestinal ailments were the most frequently treated using medicinal plant followed by sensory-neuron diseases. In similar fashion, disorders of the gastrointestinal system and parasitic infections were the commonly treated ailments and similar results have been reported in Kenya and Zegie peninsula [33,77,82]. Stomachache and diseases related to digestive system could be attributed to poor sanitation as a result of high levels of poverty within the study region as in the case cited by other studies [47,77,78,83]. High FL of a species in the scope of this study shows the extensive use of a particular plant species to treat certain diseases by the inhabitants because of its ease of accessibility and its effectiveness to treat the diseases. Such information may lead to the efficacy of these plants and their chemical and pharmacological components of the reported activity against various diseases. For example, *Carissa spinarum* L. with FL = 90.5 is good for treating malaria. The plant contains important bioactive constituents including glycosides, acids, saponins, tannins, terpenoids, and alkaloids which have medicinal value [79,84]. Within the area studied, *Carissa spinarum* L. is mainly used in treating malaria, chest pains, epilepsy, diarrhea, coughs, breast cancer, arthritis, and gonorrhea.

Different ecological climatic conditions have been characterized with different plant diversity, hence pointing to some of the probable reasons for similarities and differences of plants of medicinal value found in our study area and other extents. Data collected and analyzed from the region of study reveal remarkable differences in parts of the plant used, preparation mode of herbal medicine and their use as has been documented in other regions. However, 82% of medicinal applications were new and unique to the study at hand.

The use of medicinal plant species recorded in the same study area had 14%, 16%, and 18% JCS respectively and the neighboring areas like Kakamega [44,46,47] showed remarkable similarity at, 12%, 12%, and 13% JCS respectively. There was close resemblance in terms of the usage of medicinal plants because of close proximity to the research area.

Threatened medicinal plant species recorded in this study include *Warburgia ugandensis* Sprague. (VU) and *Ansellia africana* Lindl. (VU). The rest of the medicinal plants are either data deficient (DD), not evaluated (NE), or of least concern (LC), by IUCN. Many factors have been associated with the dangers faced by the medicinal plants in the study region. Informants’ insights show that the main threats to plants of medicinal value were forest encroachment for agricultural expansion, overharvesting, overgrazing, and environmental degradation (Table 6). Majority of the respondents indicated that agricultural activity (38.5%) was a significant danger to the existence of medicinal plants and their conservation because of an increase in human population. Some respondents pointed out that medicinal plants within the area of study had other threats as a result of deforestation and loss of habitat. A study by Mutwiwa [51] also listed overgrazing, charcoal burning, and environmental degradation as some of the threats faced by the plants of medicinal value in Machakos County, Mwala sub-county, Kenya. From our assessment in the course the field investigations, it was further noted that none of the listed medicinal plants were cultivated by the communities.

## 5. Conclusions

Planting fast-growing plant species for the production of charcoal would greatly help in lowering the harvesting of medicinal plants and enhance the conservation of vulnerable plant species. Proper grazing management of domestic animals should be enforced by the authorities in the forest reserves to reduce overgrazing especially in the areas of the forest that are more susceptible to overgrazing like the forest periphery. Proper harvesting regulations should be implemented and followed to reduce overharvesting and overexploitation of medicinal plants especially those species that are used more frequently to treat common ailments. This study provides a detailed report and an appreciation of medicinal knowledge among the three communities around the Cherangani Hills. The awareness of the importance of medicinal plants in human healthcare is important as scientific evaluation promises their future use in the development of new drugs for emerging diseases. The information on medicinal plants, dosages, and the ailments treated might be heavily eroded in the days to come because of the observed poor record keeping and the increasing use of western medication. This inventory therefore can be used as a source of information for the conservation agencies to enable proper management of plant biodiversity and its resources.

## Figures and Tables

**Figure 1 plants-09-00331-f001:**
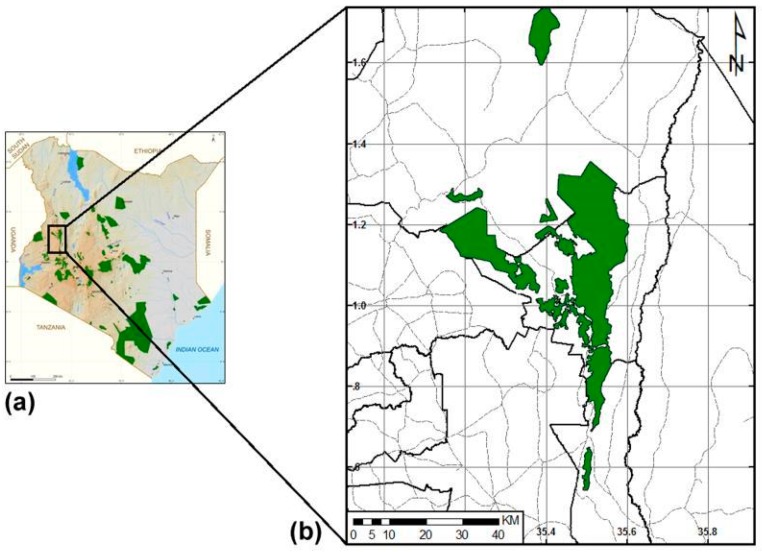
Cherangani hills forest Ecosystem. (**a**) Map of Kenya (**b**) the distribution of Cherangani hills.

**Figure 2 plants-09-00331-f002:**
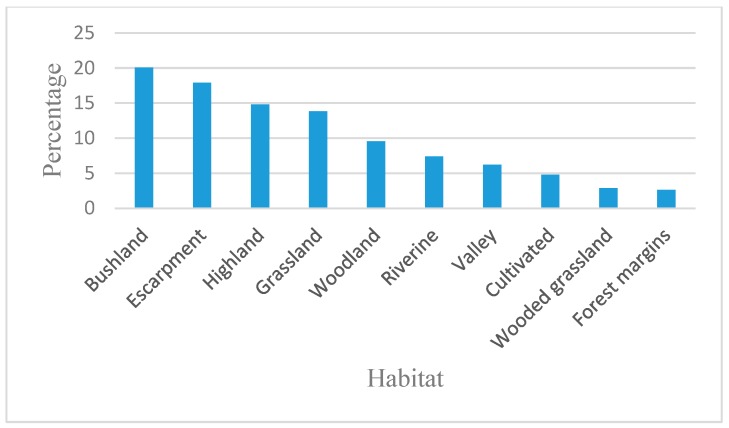
Plant habitats for medicinal plants of Cherangani Hills.

**Figure 3 plants-09-00331-f003:**
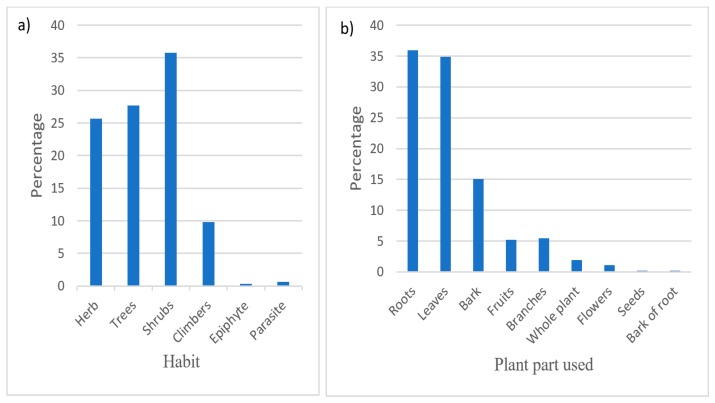
(**a**) Medicinal plant habit, (**b**) plant parts for herbal preparation around Cherangani Hills.

**Figure 4 plants-09-00331-f004:**
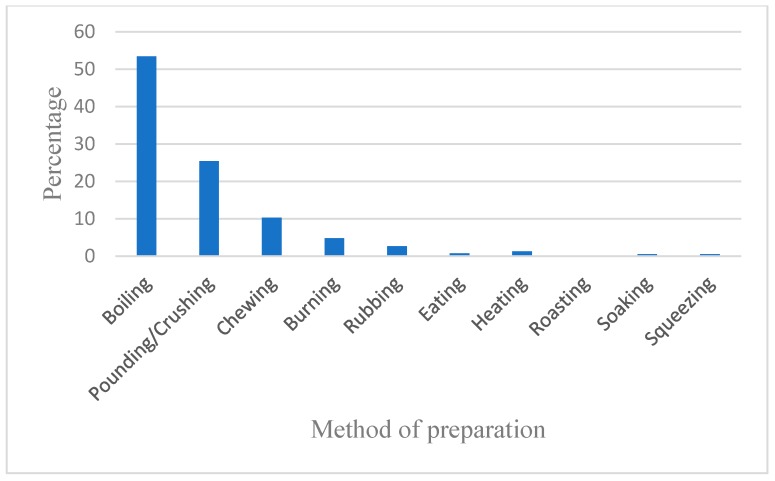
Preparation methods for medicinal plants.

**Table 1 plants-09-00331-t001:** Demographic data of the informants around Cherangani Hills.

	Count	%	Expected Mean Observation	Statistics
Gender				*p* value 0.820847
Male	40	51.28	39	
Female	38	48.72	39	
Age *		%		*p* value < 0.001
15–25	1	1.28	13	
26–30	3	3.85	13	
36–45	5	6.41	13	
46–55	10	12.82	13	
56–65	15	19.23	13	
66–75	24	30.77	13	
76–85	20	25.64	13	
Educational status *		%		*p* value < 0.001
illiterate	33	42.31	19.5	
primary	26	33.33	19.5	
secondary	17	21.79	19.5	
tertiary	2	2.7	19.5	

* Significant difference (*p* < 0.05) between the averages of paired categories.

**Table 2 plants-09-00331-t002:** Highest families and genera of medicinal plants.

Family	Species	Genera	Genera	Family	Species
Asteraceae	32	21	Acacia	Leguminosae	7
Leguminosae	28	15	Vernonia	Vernonia	6
Lamiaceae	18	11	Crotalaria	Leguminosae	5
Rubiaceae	14	9	Rhus	Anacardiaceae	5
Euphorbiaceae	12	8	Maytenus	Celastraceae	5
Apocynaceae	10	8	Solanum	Solanaceae	5
Malvaceae	10	6	Helichrysum	Asteraceae	4
Anacardiaceae	8	4	Dombeya	Malvaceae	4
Amaranthaceae	6	4	Ficus	Moraceae	4
Celastraceae	6	2	Polygonum	Polygonaceae	4
Solanaceae	6	2			

**Table 3 plants-09-00331-t003:** Informant consensus factors for categorized ailments.

Ailment Categories	Specific Conditions	Number of Used Reports	% of Total Species	No. of Taxa	ICF
Digestive system disorders	Ulcers, diarrhea, stomachache, dysentery, constipation, low appetite, nausea, purgative, intestinal worms, gastrointestinal disorders, amoeba.	139	25.18	108	0.22
Respiratory tract infections	Cold, cough, respiratory infections, asthma, bronchitis, flue, sore throat, tuberculosis	101	18.30	74	0.27
Parasitic diseases and other infections	Malaria, fever, measles, headache, yellow fever, ear, conjunctivitis, toothache, mouth blisters eye infections	99	17.93	61	0.62
Erectile dysfunctions and importance	Male sexual vitality, aphrodisiac	12	7.07	10	0.18
Gynecological issues	Fertility enhancer, heavy menstrual flows, uterine cleansing, weakness during pregnancy, induction of labor, sterility in women, induce pregnancy, removing placenta, regulation of monthly periods, abortion, after birth pains, menstrual pains	31	5.98	8	0.76
Skin infections	Wounds, burns, smallpox, ringworms, warts, skin rashes, leprosy, astringent, boils	33	5.62	9	0.75
Circulatory system diseases	Hypertension, anemia, cuts, hemorrhoids, blood cleanser, hemorrhage, heart attack reduce bleeding, edema.	22	3.99	8	0.67
Blood and Urinary system disorders	Urinary infections, kidney inflammations.	11	3.8	4	0.70
Poisonous and animal bites	Snake, centipede and insect bites	16	2.89	8	0.53
Muscular-skeletal problems inflammation	Backache, joint pains, rheumatism, fractures, joints inflammation, swollen body parts.	21	2.72	10	0.50
Neurological and nervous System disorders	Convulsions, epilepsy, memory and neurological disorders, madness reduction.	5	2.17	4	0.50
Genital apparatus diseases	Genital organs infection, sterility, infertility, prostate infections, syphilis, and gonorrhea	39	1.99	12	0.71
Metabolic disorders	Liver diseases, hepatic.	15	1.45	6	0.79
Cancers	Breast cancer, prostate cancer, skin cancer	8	0.91	4	0.57

**Table 4 plants-09-00331-t004:** Medicinal plants highly utilized in Cherangani.

Frequently Used Species	Local Name	Part Used	Popular Use	N_P_	N	FL%	References
*Sclerocarya birrea* (A.Rich.) Hochst.	Arolwa (M), Roluwo (P)	R, B	Enlarged spleen and liver	32	59	54.3	[15]
*Carissa spinarum* L.	Loketetwo (P) Eshikata (L)	R, L	Malaria	38	42	90.5	[15,50,54,67]
*Clerodendrum myricoides* (Hochst.) R.Br. ex Vatke	Chebobet (M), Shikuma (L)	R	Chest pains	16	25	64.0	[34,36,39,40,42,47,61]
*Aloe volkensii* Engl.	Cherotwo (M), Tolkos (P), Linakha (L)	L	Pneumonia	24	27	88.9	[11,15,30,44,47,68,69]
*Mondia whitei* (Hook.f.) Skeels	Mukombelo (L)	R	Aphrodisiac	24	36	66.7	[15,22,61]
*Toddalia asiatica* (L.) Lam.	Kipkeres (M), Katamwa (P)	R, Fr, B	Coughs, Colds	35	47	74.5	[15,18,36,38,40,43,46,70,71,72]
*Syzygium guineense* (Willd.) DC.	Lamaiwo (M), Cheptimanwa (P)	B, Fr	Abdominal pains	13	17	76.5	[40,44,47,70,72]
*Ricinus communis* L.	Kimonwo (M), Pondon (P) Libono (L)	R, L	Diarrhea	20	28	71.4	[11,15,36,40,46,47,51,61,71]
*Erythrina abyssinica* DC.	Gorgorwa (P), Korkorwo (M) Omurembe (L)	B, L	Indigestion	23	34	67.6	[11,15,18,34,36,40,42,46,47,61,73]
*Prunus africana* (Hook.f.) Kalkman	Tendwo (M)	B, L	Prostate cancer	9	14	64.3	[11,15,34,36,40,43,44,61,70,72]
*Asparagus racemosus* Willd.	Kabungai (M)	R	Kidney diseases	17	47	36.2	[18,36,39,40,68]
*Warburgia ugandensis* Sprague	Sokwo (M)	B, L	Respiratory disorders	20	22	90.9	[15,32]
*Withania somnifera* (L.) Dunal	Tarkukai (M), Akakagh (P)	R, L	Relives labor pains	21	29	72.4	[15,18,34,36,42,67,71,72]
*Basella alba* L.	Inderema (L),	L	Regulates monthly periods	24	28	85.7	[15,34,36,39,40,61,71,73]
*Clausena anisata* (Willd.) Hook.f. ex Benth.	Cheboinoiywa (M), Kisimbari (L)	R, B, Br	Heart diseases	6	10	60.0	[15,18,36,46,67,68]
*Periploca linearifolia* Quart.-Dill. and A.Rich.	Sinindet (M), Sinendet (P)	Br, L	Syphilis and Gonorrhea	32	43	74.4	[11,15,18,36,38,40]
*Urtica massaica* Mildbr.	Kimelei (M)	R, L	Ulcers	27	31	87.0	[15,40,44,71]
*Tragia brevipes* Pax	Kimelei (M), Chemelei (P)	Br, L, R	Rheumatism	5	13	38.5	[15,18,34,40,42,44,67,71]

Key: Local name: Marakwet = (M), Pokot = (P), Luhya = (L); Parts used (PU): L—leaves, R—roots, B—bark, Fr—fruit, Br—branches; N_P_ = represents the number of people mentioning a particular disease treated by a particular plant; N = represents the informants who used the local plants as a medicine group; FL = fidelity level.

**Table 5 plants-09-00331-t005:** A comparison of medicinal plants within the study area and those in other extents.

Study Area (County)	Year of Study	Species No. (x and y)	Common Species (z)	Jaccard’s Coefficient	% Similarity	References
Cherangani	2019	286				This review
Machakos	2018	51	23	0.06	6	[51]
Kakamega	2018	250	66	0.13	13	[46]
Kakamega	2018	94	54	0.12	12	[47]
Sungurur	2017	99	72	0.16	16	[70]
Makueni	2017	42	21	0.06	6	[35]
Nandi	2015	56	34	0.09	9	[40]
Tharaka Nithi	2015	72	21	0.06	6	[42]
Marakwet	1978	111	86	0.18	18	[18]
Kakamega	2014	65	25	0.07	7	[44]
Keiyo	2014	73	59	0.14	14	[49]
Mt. Elgon	2010	107	52	0.12	12	[36]
Nandi	2008	40	19	0.05	5	[38]
Embu and Mbeere	2007	86	45	0.11	11	[34]

**Table 6 plants-09-00331-t006:** Threats to medicinal plants in Cherangani Hills.

Threats	Frequency (N = 78)	Percentage (%)
Agricultural expansion	30	38.5
Overgrazing	16	20.5
Overharvesting	14	17.9
Firewood and Charcoal production	8	10.3
Environmental degradation	6	7.7
Others	4	5.1

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
