# Peer review of "Medicinal Plants and Their Traditional Uses in Local Communities around Cherangani Hills, Western Kenya"

_plants, 2020, doi:10.3390/plants9030331_

Round 1
Reviewer 1 Report
Dear Authors,
I believe you have done very good job in systematization of actual data on medicinal plants in three local communities at the Western Kenya. Presented results provide an obvious advance in current knowledge and would be significant for environmental and life quality issues. At the same time there are still needs to improve the quality of their presentation and interpretation.
First of all it makes sense to improve the introduction quality with increasing its general analytical aspects taking attention on the biological resources of the medicinal plants and/or estimation of their abundance, their limiting environmental factors and regional features of the investigated communities too.
You have to be more precise and clear in your results presentation too.
For example, I could not understand what means 74 years as average age (in the line 137) according to data from Table 1.
I could not understand why the total sum in age groups distribution is less than 100% in the Table 1.
I believe the title of the Table 2 could be more correct too as the total number of genera is less than 11 in it.
I’m so sorry but I could not understand the real conclusion in the sentence, Lines 212-214.
There are some technical problems too, in particular with text size in the Figure 1, gaps in Lines 6, 71, absence of the Table 6 after reference to it in Line 266.
Discussion content has some significant repetitions after previous chapter with results presentation.
On the contrary, essential part of conclusion content is not clear supported by presented results and previous discussion.
I believe you will be able to improve essentially your manuscript after these kind revisions.
Reviewer 2 Report
The paper is very interesting. Medicinal plants are very important in this area. Very well written paper; however, it has many limitations. The document is very appropriate for Plants journal.
Some suggestions:
1. The names of authors must be well written. See line 6 'and'.
2. 'References' section is not correct. See 'Guide for Authors'.
3. I recommend improving 'introduction' section.
4. I recommend including a specific section on limitations in 'discussion' section.
5. In Table 1 (line 142). 'Age data' does not add up to 100%.
6. Maybe, the conclusions could have improved with more advanced statistics. A 'multiple correspondence analysis' could have been very useful.
